

# Identification of the occurrence and potential mechanisms of heterotopic ossification associated with 17-beta-estradiol targeting MKX by bioinformatics analysis and cellular experiments

Yunpeng Zhang[1,*], Jingwei Zhang[2,3,*], Chenyu Sun[4] and Fan Wu[1]

[1] Department of surgery, Shanghai Fengxian District Central Hospital, Shanghai, China
[2] Department of Orthopedics, Shanghai Fengxian District Central Hospital, Shanghai, China
[3] Shanghai University of Medicine & Health Sciences Affiliated Sixth People's Hospital South Campus, Shanghai, China
[4] Department of Internal Medicine, AMITA Health Saint Joseph Hospital Chicago, Chicago, Illinois, United States of America
* These authors contributed equally to this work.

Corresponding author
Fan Wu, WF131@seu.edu.mk

## ABSTRACT

**Background:** Tendon heterotopic ossification (HO) is a common condition occurring secondary to tendon injury or surgical trauma that significantly affects the patient's quality of life. The treatment of tendon HO remains challenging due to a lack of clarity regarding the pathological mechanism. Mohawk (MKX) is a key factor in preventing tendon HO; however, its upstream regulatory mechanism remains to be understood. This study aimed to identify potential compounds that target and regulate MKX and explore their functional mechanisms.

**Methods:** Bioinformatics analysis of MKX-related compounds and proteins was performed based on data from the STITCH and OncoBinder databases. Subsequently, the SymMap database was used to study MKX-related traditional Chinese medicine drugs and symptoms. Next, the OncoBinder genomic and proteomic discovery model was applied to identify potential regulators of MKX. The analytical tool Expert Protein Analysis System for proteomics was used to predict the three-dimensional structure of MKX, and the AutoDockTools software was used to identify pockets of activity at potential sites for molecular docking. Furthermore, we evaluated the effect of different doses of 17-beta-estradiol on bone marrow-derived mesenchymal stem cells (BM-MSCs).

**Results:** By predicting the three-dimensional structure of MKX and simulating molecular docking, Pro-Tyr and 17-beta-Estradiol were found to target and bind to MKX. Analysis of the STITCH and OncoBinder databases showed that MKX had a significant regulatory correlation with suppressor interacting 3 A/histone deacetylase 1 (SIN3A/HDAC1). The GO and KEGG pathway enrichment analysis revealed that the functions of MKX and its associated proteins were mainly enriched in osteogenic-related pathways. Assessment of the proliferation of BM-MSCs revealed that 17-beta-estradiol possibly upregulated the mRNA expression of the HDAC1-SIN3A/BMP pathway-related RUNX2, thereby promoting the proliferation of BM-MSCs.

**Conclusions:** The compounds Pro-Tyr and 17-beta-Estradiol may bind to MKX and thus affect the interaction of MKX with SIN3A/HDAC1.

# INTRODUCTION

Tendons are connective tissues that connect muscles and bones. Tendons play a role in transmitting muscle contractions to the bone and are highly susceptible to injury during exercise (*Giddings et al., 2000*). Statistics reveal that more than 16 million people in the United States suffer from key muscle injuries each year (*James et al., 2008*). Tendon heterotopic ossification (HO) is a common condition occurring secondary to tendon injury or surgical trauma, which can lead to varying degrees of pain, edema, decreased mechanical properties of the tendon or even rupture, and limited joint movement that can significantly affect the patient's quality of life. The treatment of tendon HO remains a major challenge due to a lack of clarity regarding the pathological mechanism (*O'Brien et al., 2012*). Therefore, a better understanding of the pathological mechanism will provide new therapeutic targets and ideas for the treatment of tendon HO.

Recent studies have shown that knockdown of the tendon-specific transcription factor Mohawk (MKX) leads to ectopic tendon ossification, suggesting that MKX may be a key factor in preventing the ossification (*Suzuki et al., 2016*; *Liu, Xu & Jiang, 2019*). MKX is essential for the development and differentiation of tendons (*Ito et al., 2010*; *Liu et al., 2010*, *2015*; *Kimura et al., 2011*; *Suzuki et al., 2016*). *In vitro* culture of MKX-knockdown rat tendon-derived cells and induction of differentiation showed that the knockdown accelerated chondrogenesis and osteogenic differentiation, while overexpression of MKX inhibited chondrogenesis, osteogenesis, and adipogenic differentiation (*Suzuki et al., 2016*). These results suggest that MKX has a dual function, namely maintaining the normal differentiation of tendon cells and inhibiting the expression of chondrogenesis-related genes, thus suppressing the transdifferentiation of tendon and ligament cells to other lineages. Currently, there is limited understanding of the upstream regulatory mechanism of MKX.

In the STITCH database, each compound-protein interaction corresponds to a score indicating the probability of interaction and affinity between them (*Szklarczyk et al., 2016*). We used data from the STITCH (http://stitch.embl.de) database to identify MKX-related compounds and proteins. SymMap database was used in this study to examine the MKX-related herbal drugs and symptoms (*Wu et al., 2019*); OncoBinder genomic and proteomic discovery models were used to identify potential regulators of MKX (*Van Coillie et al., 2016*). The Expert Protein Analysis System (ExPASy) is a specialized proteomics analysis tool that combines information from several databases, including SWISS-PROT, TrEMBL, ENZYME, PROSITE, and the SWISS-MODEL repository (*Gasteiger, 2003*). ExPASy was used to predict the structure of MKX in this study, and AutoDockTools software was used for molecular docking. The effects of

different doses of 17-β-estradiol on bone marrow mesenchymal stem cells (BM-MSCs) were also evaluated.

We speculated that MKX could maintain normal differentiation of tendon cells and inhibit the expression of genes related to cartilage formation. Therefore, this study aimed to identify potential molecules that target and regulate MKX, reveal the mechanism by which MKX contributes to the development of HO, identify novel effective therapeutic targets, and improve the diagnosis and prognosis of patients with HO.

## MATERIALS AND METHODS

### Bioinformatics analysis of MKX and Structure prediction of MKX

We used data from the STITCH (http://stitch.embl.de) database to identify MKX-related compounds and proteins (*Szklarczyk et al., 2016*). We used data from the SymMap (http://www.symmap.org/) database to identify MKX-related traditional Chinese medicine (TCM) drugs and symptoms (*Wu et al., 2019*). We also applied a genomic and proteomic discovery model named OncoBinder to identify potential regulators of MKX (*Van Coillie et al., 2016*). The tool was used to predict the structure of MKX (*Gasteiger, 2003*).

### Enrichment analysis

The GO and KEGG pathway enrichment analysis was performed as described in the citation and visualized by R software (*Yu et al., 2012*). The clusterProfiler package is used for enrichment analysis, while the org.Hs.eg.db package is used for ID conversion. Gene Set Enrichment Analysis (GSEA) analysis was performed using the clusterProfiler package of R software [version 3.14.3] (*Subramanian et al., 2005*). The reference gene collection used by GSEA is c2.cp.v7.2.symbols.gmt, and the gene set database used is MSigDB collections. Conditions were considered significantly enriched when the false discovery rate was < 0.25 and p.adjust was < 0.05.

### Molecular docking analysis

Molecular docking analysis is an important method for predicting compound and protein binding (*Friesner et al., 2004*; *He et al., 2019*; *Mehta, Miszta & Filipek, 2021*). The structures of Pro-Tyr and 17-beta-estradiol were searched in the PubChem database. After the three-dimensional structure of MKX was predicted by the ExPASy tool, the AutoDockTools software was used to determine the active pockets of potential sites for molecular docking. The structure has been verified and structure minimization has been performed before the docking study. Finally, the AutoDock Vina software was used to perform molecular docking to verify the targeting of MKX with Pro-Tyr and 17-beta-estradiol. The lowest binding energy values were considered to visualize the results using the PyMOL software.

### Molecule dynamics

The complex system of each compound with the target egg obtained by docking was used as the initial structure to perform all-atom molecular dynamics simulations (MD)

separately. The charges of the small molecules were calculated using the antechamber module and the Hartree-Fock (HF) SCF/6-31G* of the gaussian 09 software. Afterwards, small molecules and proteins were described using force fields (*Wang et al., 2004*; *Maier et al., 2015*). The LEaP module was used to add hydrogen atoms to the system, a truncated tetrahedral TIP3P solvent cartridge was added at a distance of 10 Å from the system, and Na+/Cl- was added to the system for balancing the system charge.

Molecular dynamics simulations were performed using AMBER 18 software (*Lee et al., 2020*). After the energy optimization of the system was completed, a 200 ps warming of the system at a fixed volume and constant warming rate was used to slowly increase the system temperature from 0 K to 298.15 K. Finally, these composite systems are simulated with 30 ns NPT (isothermal isobaric) tethering under periodic boundary conditions, respectively.For the simulations, the truncation distance of the non-bond was set to 8 Å. The Particle mesh Ewald (PME) method was used to calculate the long-range electrostatic interaction, the SHAKE method was used to limit the bond length of hydrogen atoms, and the Langevin algorithm was used for temperature control (*Larini, Mannella & Leporini, 2007*; *Mehta, Miszta & Filipek, 2021*). Traces were saved at 10 ps intervals for subsequent analysis.

## MMGBSA free energy calculation

The binding free energy between protein and ligand for all systems was calculated by the MM/GBSA method (*Genheden & Ryde, 2015*; *Chen et al., 2020*). The MD trajectory of 25–30 ns was used as the calculation in this study. And the specific equation was as follows:

$$\Delta G_{bind} = \Delta G_{complex} - \left(\Delta G_{receptor} + \Delta G_{ligand}\right)$$
$$= \Delta E_{internal} + \Delta E_{VDW} + \Delta E_{elec} + \Delta G_{GB} + \Delta G_{SA}$$

$\Delta E_{internal}$ represents internal energy, $\Delta E_{VDW}$ represents van der Waals interaction, and $\Delta E_{elec}$ represents electrostatic interaction. $\Delta G_{GB}$ and $\Delta G_{GA}$ were collectively referred to as the solvation free energy. Where GGB was the polar solvation free energy and GSA was the nonpolar solvation free energy. The GB model developed by Nguyen was used to calculate $\Delta G_{GB}$ (*Nguyen, Roe & Simmerling, 2013*).

## Experimental cells and culture

The cells used in this study were human, primary bone marrow-derived mesenchymal stem cells (BM-MSCs, Normal, Human, ATCC PCS-500-012), purchased from the American Type Culture Collection (ATCC). BM-MSCs were used, which was supplemented with 7% Fetal Bovine Serum (FBS), 15 ng/mL insulin-like growth factor 1 (rhIGF-1), 125 pg/mL fibroblast growth factor-basic (rhFGF), 2.4 mM L-alanyl-L-glutamine, and 1% penicillin-streptomycin solution (P/S). All cells were cultured at 37 °C in a cell culture incubator with 5% $CO_2$, and the experiment was completed with five passages of cells. Passaged BM-MSCs ($n = 1,000$) were placed in each well of a 96-well plate, divided into three groups, and were exposed to different doses of 17-beta-estradiol (0, 0.001, 0.01 nmol/L). Accordingly, the groups were categorized as control, 0.001 nmol/L 17-beta-estradiol-treated, and 0.01 nmol/L 17-beta-estradiol-treated.

### Real-time quantitative reverse transcription PCR

Real-time quantitative reverse tanscription polymerase chain reaction (RT-qPCR) was performed to determine the mRNA expression of runt-related transcription factor 2 (RUNX2). RT-PCR was performed to quantify the expression of genes and cytokines associated with RUNX2 at 1, 3, and 5 d after the induction. Total RNA was extracted from the BM-MSCs using the total RNA extraction reagent (TRIzol) and purified using the QIAquick PCR purification kit. The first-strand cDNA synthesis kit and random hexamer primers were used for the reverse transcription reaction to obtain the cDNA using DNAse I. The quantitative RT-PCR was performed using a fluorescent quantitative PCR kit (SYBR Green PCR Master Mix). β-actin was selected as the internal reference (Table S1). PCR cycling condition was adjusted as follows: 95 °C for 2 min, 94 °C for 30 s, 55 °C for 30 s, and then at 72 °C for 1 min with 40 cycles performed. 2-DDCt method was used to obtain fold expression results.

### Cell viability assay

Cell viability and proliferation were measured using the Cell Proliferation Kit I (MTT; Roche, Basel, Switzerland) according to the manufacturer's instructions and the method described in previous studies (*Jiang et al., 2020*; *Zhao et al., 2021*, p. 5). First, the transfected cells were stabilized by trypsin digestion. Following this, the cells were centrifuged, collected, resuspended in a single-cell suspension, and seeded in 96-well plates at the density of $1 \times 10^4$ cells/well. The cell cultures were maintained. A multifunctional microplate reader was used to measure optical density (OD) values. Cell growth curves were plotted using OD as the vertical axis and time as the horizontal axis.

### Statistics and analysis

Statistical and visual analyses were performed using R software [version 3.6.0]. The measurement data were expressed as x ± s and compared using the t-test; the count data were expressed as percentages and compared using the χ2 test. $p < 0.05$ was considered statistically significant.

## RESULTS

### Construction of a regulatory network for MKX, related genes and compounds based on the STITCH database

We used data from the STITCH database (http://stitch.embl.de) to identify MKX-related compounds and proteins and constructed MKX-related gene and compound regulatory networks (Fig. 1A). Compounds related to MKX are Pro-Try and arsenite. Proteins related to MKX include SCXA, PROX2, suppressor interacting 3 A/histone deacetylase 1 (SIN3A/HDAC1), PBX1, MYOD1, TCF15, and PBX4. The regulatory network has nine nodes and 13 edges. Based on these proteins, we completed the GO functional enrichment analysis. The signaling pathways in which the MKX-related proteins were found to be enriched are as Fig. 1B. These results suggest that MKX-related protein functions are enriched in biological processes of biological development and transcriptional regulation.
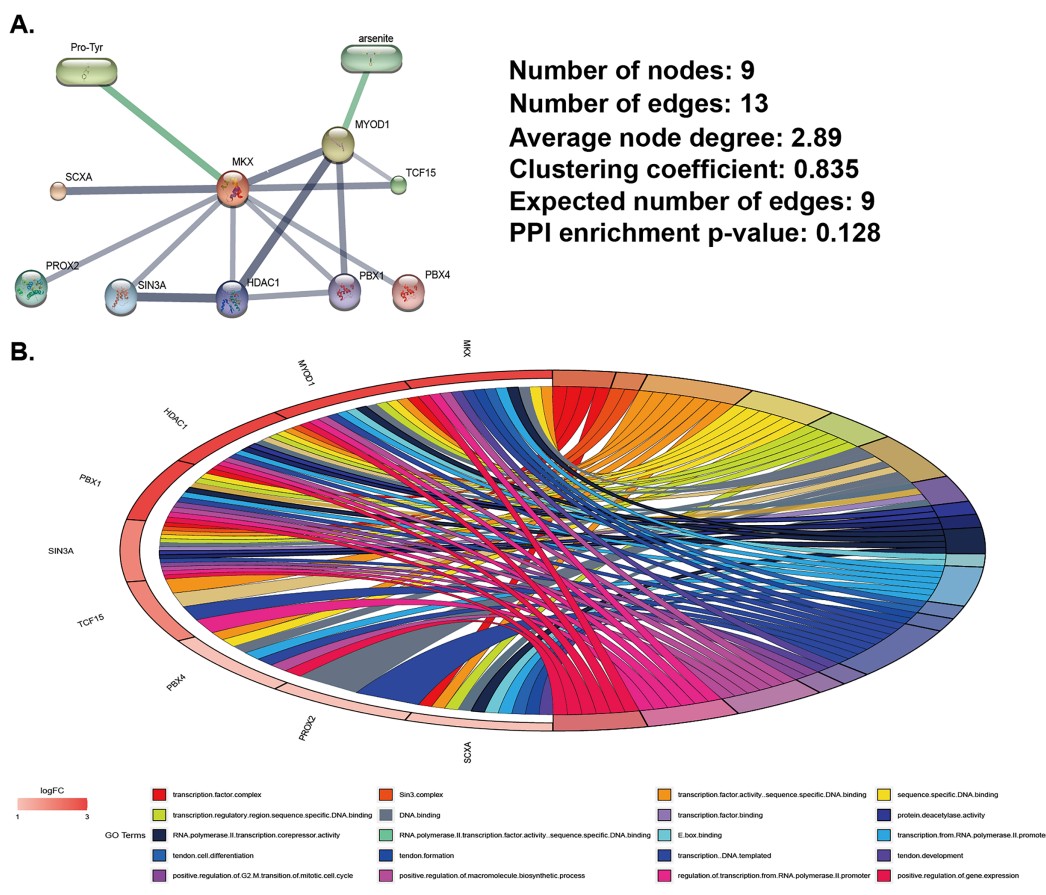

**Figure 1 Analysis of MKX-related gene and compound regulatory networks.** (A) Construction of MKX, related genes, and compound regulatory networks based on data from the STITCH database; (B) the GO function analysis of genes in the subnetwork.

## MKX-related symptom network constructed using SymMap

The SymMap database integrates TCM and MM data on drug treatments and their molecular mechanisms, as shown in Fig. 2. Herbs, ingredients of active compounds, and TCM and MM symptoms are included in this network. The compound 17-beta-estradiol, which can be derived from three herbs (*Moschus, Cervi cornu Pantotrichum, and Armeniacae Semen Amarum*), was found to be capable of targeting and binding to MKX. MKX was found to be potentially associated with nine diseases, which in turn were associated with 42 symptoms (Table 1). In summary, the network demonstrates both TCM and MM symptoms associated with MKX.

## Molecular docking validation of the compounds Pro-Tyr and 17-beta-estradiol for target binding of MKX

The predicted three-dimensional structure of MKX based on the ExPASy tool is shown in Fig. 3A. The structures of Pro-Tyr and 17-beta-estradiol that were obtained from the PubChem database are shown in Fig. 3B. Molecular docking using the AutoDock Vina software revealed that Pro-Tyr and 17-beta-estradiol target binding of MKX were at different loci (Figs. 3C and 3D).

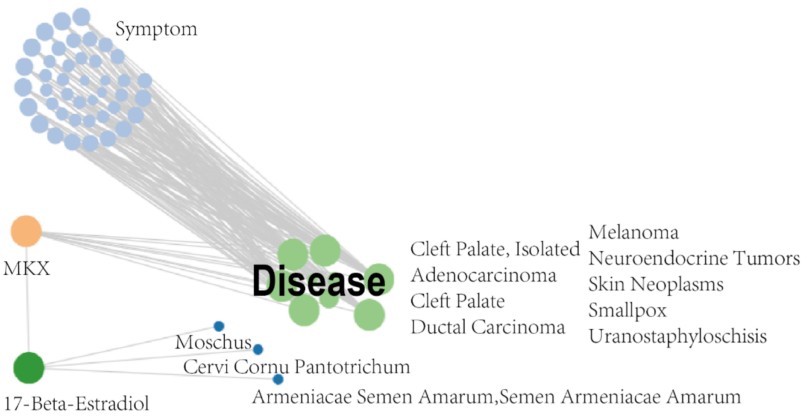

**Figure 2 MKX-related symptom network built with SymMap.** MKX-related symptom network built with SymMap.

## Stability of complexes and molecular dynamics analysis

Pro-Tyr as well as Estradiol fluctuated less during the kinetic simulations, indicating that they were both stably bound within the active pocket (Figs. 4A, 4B). RMSF showed that Pro-Tyr could better reduce the flexibility of the protein, which implied that Pro-Tyr was more stable with the protein system (Fig. 4C). As shown in Figs. 5A and 5B, both molecules were maintained at the same binding site after MD simulation analysis. The small molecule seems to be closer to the protein after MD simulation analysis, which indicates that both molecules bind stably to the target protein. The results of MM/GBSA calculations showed that Estradiol/protein binding free energy was −17.57 ± 2.97 kcal/mol and Pro-Tyr/Protein binding free energy was −13.79 ± 2.68 kcal/mol (Table 2). The binding energy of Estradiol/protein is mainly contributed by van der Waals interaction, and that of Pro-Tyr/Protein is mainly contributed by electrostatic interaction.

## Identification and functional analysis of proteins that may regulate MKX

Analysis of the OncoBinder database revealed a significant regulatory correlation between MKX and a series of proteins (TBP, HSP90A, SIN3A, TBL1X, TBL1XR1, HSP90AA1, HDAC1, ELAVL1, GTF2B, and GTF2A1) (Fig. 6A). We performed the GO and KEGG pathway enrichment analysis to identify the functions of these proteins (Fig. 6B). The GO enrichment analysis revealed that the functions of these proteins were significantly enriched in protein deacylation, protein deacetylation, histone deacetylation, transcriptional repressor complex, histone deacetylase complex, transcription factor complex, repressing transcription factor binding, RNA polymerase II general transcription initiation factor activity, and RNA polymerase II repressing transcription factor binding. The KEGG enrichment analysis revealed that these proteins were significantly enriched in viral carcinogenesis, basal transcription factors, and the IL-17 signaling pathway. The results suggest that MKX and the proteins significantly associated with it play important roles in protein acetylation and transcriptional regulation.

**Table 1  Potential symptoms related with MKX.**

| ID | Term | p value |
| --- | --- | --- |
| SMMS00054 | Constipation | $p < 0.001$ |
| SMMS00096 | Hemoptysis | 0.039 |
| SMMS00115 | Vertigo | 0.010 |
| SMMS00132 | Hoarseness | $p < 0.001$ |
| SMMS00138 | Toothache | 0.002 |
| SMMS00223 | Chest Pain | 0.008 |
| SMMS00225 | Dyspnea | $p < 0.001$ |
| SMMS00233 | Edema | 0.001 |
| SMMS00236 | Muscle Weakness | $p < 0.001$ |
| SMMS00261 | Trismus | $p < 0.001$ |
| SMMS00279 | Profound Mental Retardation | $p < 0.001$ |
| SMMS00327 | Emesis | 0.006 |
| SMMS00329 | Crescendo Angina | $p < 0.001$ |
| SMMS00332 | Jaundice | $p < 0.001$ |
| SMMS00349 | Arthralgia | $p < 0.001$ |
| SMMS00369 | Polyuria | $p < 0.001$ |
| SMMS00379 | Paralysis | 0.001 |
| SMMS00403 | Cough | $p < 0.001$ |
| SMMS00415 | Nausea | $p < 0.001$ |
| SMMS00441 | Skin Pruritus | 0.010 |
| SMMS00457 | Spasm | 0.026 |
| SMMS00461 | Paresis | $p < 0.001$ |
| SMMS00522 | Dysphonia | $p < 0.001$ |
| SMMS00540 | Angina Pectoris | 0.001 |
| SMMS00549 | Facial Paralysis | 0.006 |
| SMMS00553 | Hemiplegia | $p < 0.001$ |
| SMMS00577 | Musculoskeletal Pain | 0.002 |
| SMMS00629 | Hyperemesis Gravidarum | $p < 0.001$ |
| SMMS00633 | Cyanosis | 0.001 |
| SMMS00653 | Neuralgia | 0.002 |
| SMMS00656 | Backache | $p < 0.001$ |
| SMMS00682 | Sleep Disorder | 0.006 |
| SMMS00774 | Loss Of Appetite | 0.001 |
| SMMS00870 | Face Pain | $p < 0.001$ |
| SMMS00871 | Neck Pain | $p < 0.001$ |
| SMMS00901 | Purpura | 0.003 |
| SMMS00928 | Headache | 0.002 |
| SMMS00973 | Anorexia Symptom | 0.001 |
| SMMS01036 | Fever Symptoms | $p < 0.001$ |
| SMMS01116 | Spasticity Muscle | $p < 0.001$ |
| SMMS01125 | Thrombocytopenic Purpura | 0.001 |
| SMMS01143 | Vomiting | 0.003 |

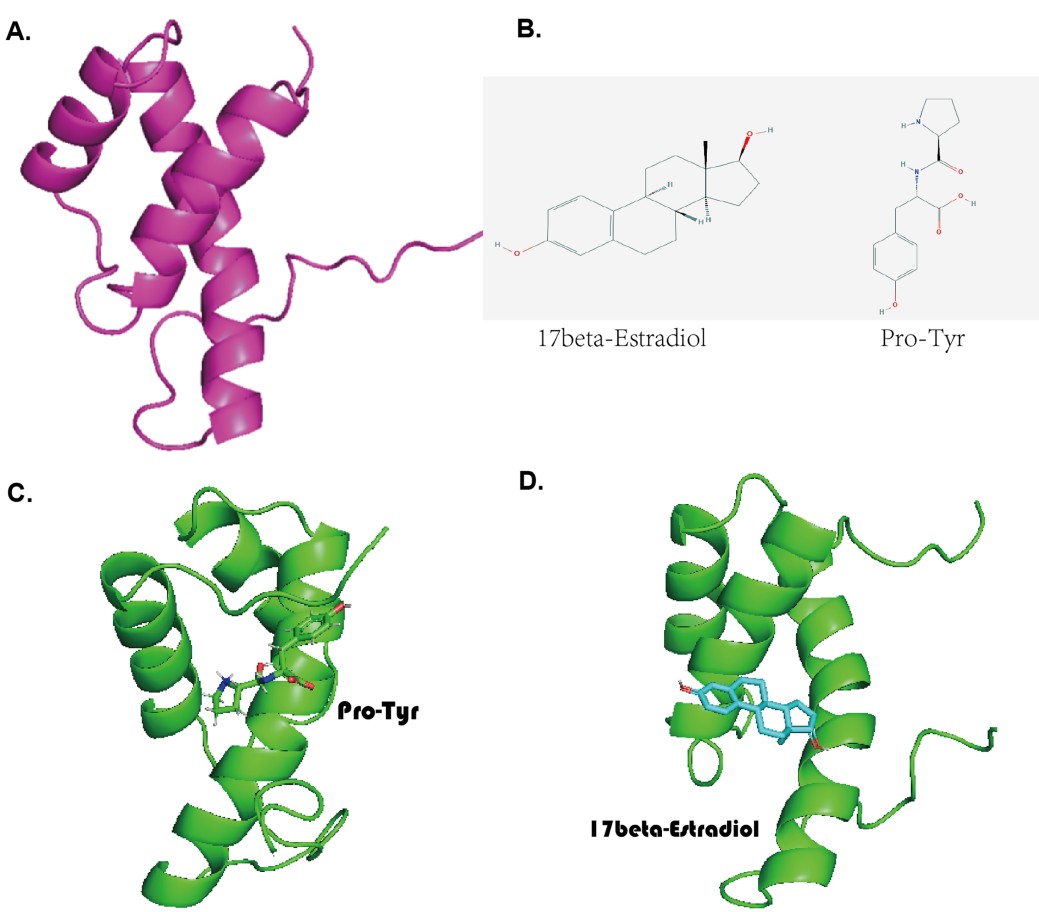

**Figure 3 Pro-Tyr and 17-beta-estradiol can target and bind MKX.** (A) The predicted three-dimensional structure of MKX based on the ExPASy tool; (B) two-dimensional structures of Pro-Tyr and 17-beta-estradiol; (C–D) schematic representation of Pro-Tyr and 17-beta-estradiol targeting MKX binding at different loci.

By combining the results presented in Figs. 1–3, it may be stated that the compounds Pro-Tyr and 17-beta-estradiol may bind MKX and affect its interaction with SIN3A and HDAC1 (Fig. 7A). The integrated GSEA based on the MSigDB collections gene set database revealed that the low expression of MKX was significantly associated with osteoblast differentiation, signaling by NOTCH4 and Hedgehog, and the Hedgehog signaling pathway enrichment demonstrated significant correlation (Fig. 7B).
We evaluated the proliferation of BM-MSCs after treatment with different doses of 17-beta-estradiol and observed an increase in the proliferation rate (Fig. 7C). This could be attributed to the upregulation of the HDAC1-SIN3A/bone morphogenetic protein (BMP) pathway-related RUNX2 mRNA expression by 17-beta-estradiol (Fig. 7D). The results of our study suggest that 17-beta-estradiol may upregulate the HDAC1-SIN3A/BMP pathway.

## DISCUSSION

By predicting the three-dimensional structure of MKX and simulating molecular docking, this study revealed that Pro-Tyr and 17-beta-estradiol could target and bind MKX.

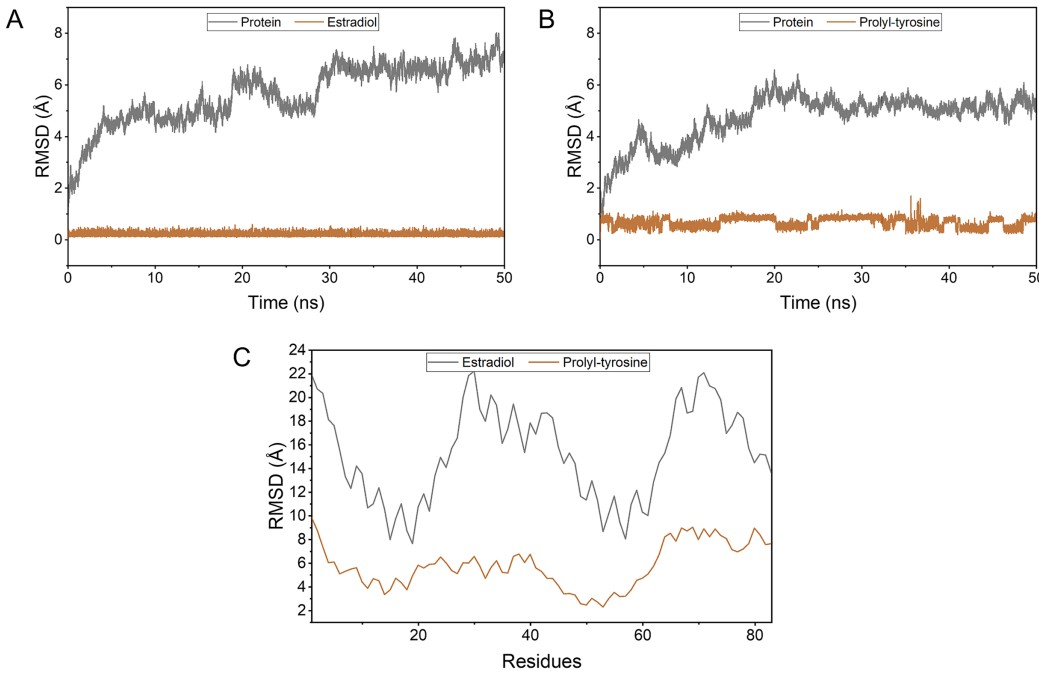

**Figure 4 The stability of the complexes as well as the binding energy.** To simulate the fluctuation of small molecules or proteins in the process, expressed as root mean square deviation (RMSD) (A, B); Changes in the flexibility of the protein in the presence of different ligand binding (C).

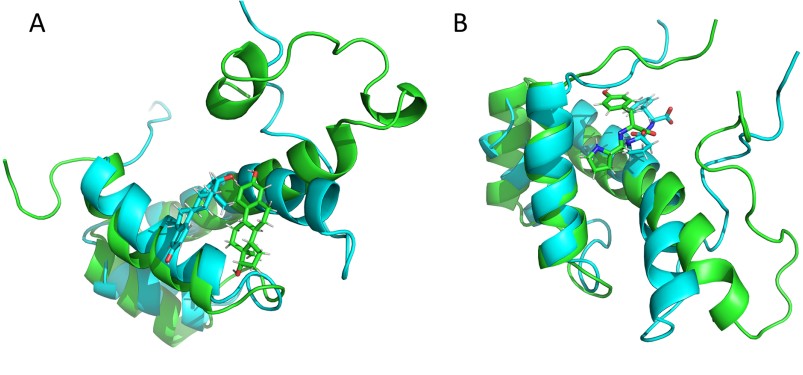

Estradiol/protein complex          Prolyl-tyrosine/protein complex

**Figure 5 Protein conformation before and after MD.** 17-beta-estradiol/MKX protein complex (A) and Pro-Tyr/MKX protein complex (B). Cyan color indicates pre-MD mimic conformation and green color indicates post-MD mimic conformation.

The present study identified potential compounds that target and regulate MKX and explore their functional mechanisms. Based on data from the STITCH and OncoBinder databases, MKX showed a significant regulatory correlation with SIN3A/HDAC1. The proliferation of BM-MSCs increased after treatment with different doses of 17-beta-estradiol. This may be attributed to the upregulation of the HDAC1-SIN3A/BMP pathway-related RUNX2 mRNA expression by 17-beta-estradiol (*Kim & Lassar,*

**Table 2 Binding free energies and energy components predicted by MM/GBSA (kcal/mol).**

| System name | Estradiol/protein | Pro-Tyr/Protein |
|---|---|---|
| $\Delta E_{vdw}$ | $-22.14 \pm 3.08$ | $-24.48 \pm 4.93$ |
| $\Delta E_{elec}$ | $-3.04 \pm 3.98$ | $-36.72 \pm 17.27$ |
| $\Delta G_{GB}$ | $10.32 \pm 4.23$ | $51.05 \pm 15.29$ |
| $\Delta G_{SA}$ | $-2.70 \pm 0.32$ | $-3.62 \pm 0.32$ |
| $\Delta G_{bind}$ | $-17.57 \pm 2.97$ | $-13.79 \pm 2.68$ |

Notes:
$\Delta E_{vdW}$, van der Waals energy.
$\Delta E_{elec}$, electrostatic energy.
$\Delta G_{GB}$, electrostatic contribution to solvation.
$\Delta G_{SA}$, non-polar contribution to solvation.
$\Delta G_{bind}$, binding free energy.

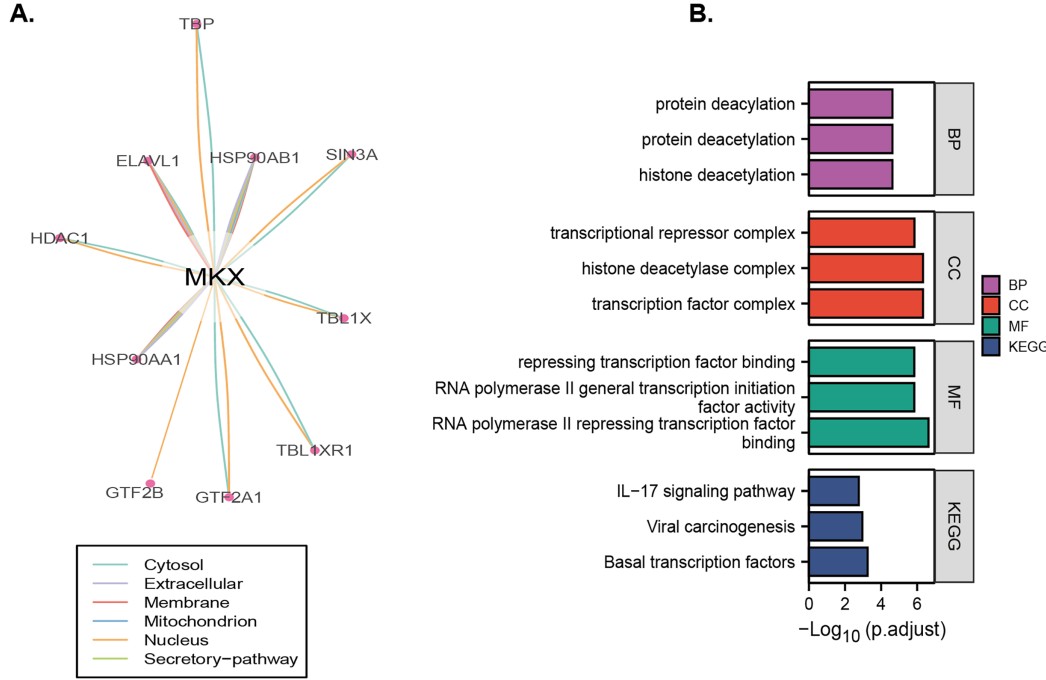

**Figure 6 Identification and enrichment analysis of proteins that may regulate MKX.** (A) The "co-activation" scores of candidate MKX-associated proteins are plotted in the lower panel, and the lines between the proteins indicate a statistical relationship between them ($p < 0.05$); (B) the GO function and KEGG pathway enrichment analysis of the MKX-related proteins. Analysis of the potential functions of MKX and 17-beta-estradiol for osteogenic differentiation.

_2003_). The results of our study suggest that 17-beta-estradiol may upregulate the HDAC1-SIN3A/BMP pathway.

As a metabolite, Pro-Tyr is a dipeptide composed of L-proline and L-tyrosine residues (_Klein et al., 1991_). As a steroidal sex hormone, estradiol is actively involved in bone metabolism and has a multifunctional regulatory role in many cell types, including BM-MSCs. A recent study showed that supplementation with 17-beta-estradiol was effective in stimulating the proliferative capacity of human and mouse BM-MSCs. In addition, 17-beta-estradiol is known to increase the differentiation potential of MSCs,

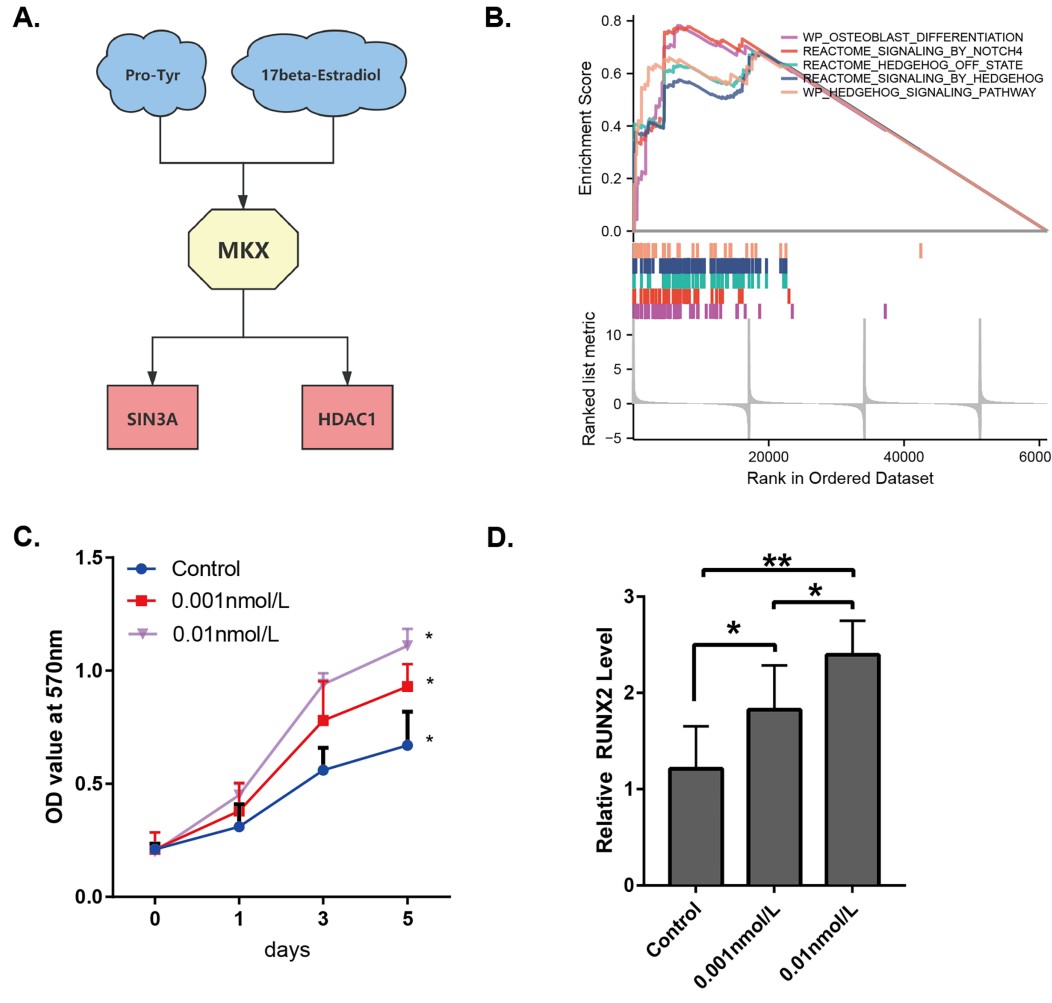

**Figure 7** **Analysis of the potential functions of MKX.** (A) Interrelationship between Pro-Tyr, 17-beta-estradiol, MKX, SIN3A, and HDAC1; (B) GSEA of MKX; (C) proliferation of BM-MSCs after treatment with different doses of 17-beta-estradiol; (D) effect of different doses of 17-beta-estradiol on RUNX2 mRNA expression.

including osteogenic and lipogenic differentiation. Furthermore, 17-beta-estradiol regulates cellular function through its action on the estrogen receptor. Estrogen has been reported to increase osteogenic differentiation of bone marrow cells by upregulating osteogenic growth factors, including BMP-2, BMP-6, and transforming growth factor-beta 1 (TGF-beta 1) (*Sammons et al., 2004*). Estradiol has been reported to enhance the BMP-induced expression of RUNX2 (*Matsumoto et al., 2010*). SIN3A serves as a support component of the chromatin repression complex SIN3/HDAC (*Zhao et al., 2019*). Previous studies have demonstrated that SIN3A promotes the transcriptional repressor activity of Nkx3.2, thereby permitting BMP signaling and induction of somatic chondrogenesis by blocking the expression of anti-chondrogenic genes (*Kim & Lassar, 2003*). HDAC is involved in regulating the differentiation of stem cells and

osteoblasts (*McGee-Lawrence & Westendorf, 2011*). Dynamic activities in the skeletal microenvironment require and contribute to rapid and temporal changes in gene expression within the cells responsible for the maintenance of bone cells (osteoblasts, osteoclasts, and osteocytes). The presence of HDAC1 in the *promoter region* of osteogenic genes is lower in differentiated osteoblasts. Numerous studies have reported that HDAC1 plays a role in the inhibition of osteoblast differentiation.

MKX, also known as Irxl (Iroquois homeobox-like 1) (*Liu et al., 2010*), is a member of the triamino-acid loop superfamily. MKX is necessary during the development of many tissues and organs in the cell proliferation, differentiation, and other processes (*Selleri et al., 2001*; *Brendolan et al., 2005*; *Moens & Selleri, 2006*; *van Tuyl et al., 2006*; *DiIorio et al., 2007*). MKX exerts a transcriptional repressive activity by recruiting the SIN3A/HDAC repressor complex (*Bilioni et al., 2005*; *Anderson et al., 2009*). In the present study, the results of GO and KEGG pathway enrichment analysis showed that MKX and its associated proteins play important functions in protein acetylation and transcriptional regulation and are closely involved in the biological processes of development and transcriptional regulation. These findings are consistent with those reported in previous studies. An integrated GSEA based on the MSigDB collections gene set database revealed that high MKX expression inhibited osteogenesis.

Based on the existing literature and the results of the present study, it is reasonable to speculate that the compounds Pro-Tyr and 17-beta-estradiol may bind MKX and competitively inhibit the binding of MKX to SIN3A/HDAC1. This could interfere with the osteogenic inhibitory effect of MKX, thereby promoting osteogenesis, which in turn manifests as HO during tendon healing.

By predicting the three-dimensional structure of MKX and simulating molecular docking, we found that Pro-Tyr and 17-beta-estradiol could target MKX. Based on data from the STITCH and OncoBinder databases, we identified a protein, SIN3A/HDAC1, with significant regulatory relevance to MKX. The results of GO and KEGG pathway enrichment analysis suggested that the functions of MKX and its associated proteins were mainly enriched in osteogenic pathways. The compounds Pro-Tyr and 17-beta-estradiol may bind MKX and affect its interaction with SIN3A/HDAC1. This study initially revealed the interaction of MKX and its associated proteins with the small molecules, Pro-Tyr and 17-beta-estradiol, and the potential mechanism of HO. It provides new targets and ideas for the future development of novel anti-HO drugs. However, the present study was limited to bioinformatics analysis, and the exact mechanism of action of MKX, Pro-Tyr, 17-beta-estradiol, and SIN3A/HDAC1 remains unclear. Future antagonist and animal model experiments are necessary to validate the mechanisms.

## CONCLUSIONS

This study initially revealed the interaction of MKX and its associated proteins with the small molecules, Pro-Tyr and 17-beta-estradiol, and the potential mechanism of HO. It provides new targets and ideas for the future development of novel anti-HO drugs.

## ACKNOWLEDGEMENTS

We would like to thank the anonymous reviewers for their valuable comments on our revisions. We thank Bullet Edits Limited for the linguistic editing and proofreading of the manuscript.

### Funding

The authors received no funding for this work.

### Competing Interests

The authors declare that they have no competing interests.

### Author Contributions

- Yunpeng Zhang conceived and designed the experiments, performed the experiments, analyzed the data, prepared figures and/or tables, authored or reviewed drafts of the paper, and approved the final draft.
- Jingwei Zhang conceived and designed the experiments, performed the experiments, analyzed the data, authored or reviewed drafts of the paper, and approved the final draft.
- Chenyu Sun analyzed the data, authored or reviewed drafts of the paper, and approved the final draft.
- Fan Wu conceived and designed the experiments, performed the experiments, analyzed the data, prepared figures and/or tables, authored or reviewed drafts of the paper, and approved the final draft.

### Data Availability

All the original data is available in the Supplemental File.

### Supplemental Information

Supplemental information for this article can be found online at http://dx.doi.org/10.7717/peerj.12696#supplemental-information.

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
