# Peer review of "Identification of the occurrence and potential mechanisms of heterotopic ossification associated with 17-beta-estradiol targeting MKX by bioinformatics analysis and cellular experiments"

_PeerJ, doi:10.7717/peerj.12696_

## Round 0.1 · original submission · Major Revisions

Dear Authors,

According to the insightful Comments of external reviewers, there are many critical concerns needed to be addressed before your manuscript can be further considered.

For example, the PCR primers must be checked again. You have to revise the Materials & Methods and the Results of the PCR analysis.

Also, the Statistical analysis method is fundamental to a study. Please clearly state how you performed the Statistical analysis.

And, the English writing needs to be improved for the revision.

Reviewer 1 ·

Basic reporting

1. Please alphabetically re-order your keywords.
2. Problem is well stated but the aim of the study should be better presented. We should be seeing your hyphotesis and why you made this study in the introduction section.
3. The used databases are being presented in the materials & methods section. Those informations should be given in the introduction.
4. The websites should be placed in the references section as internet references.
5. Please make sure you used latin alphabet symbols. There are too many round shaped symbols instead of “.” in the manuscript.
6. Please check all the spacings. There are too many spacing errors between words.
7. The presentation of databeses should be moved into the introduction section.
8. Fatal bovine serum should be corrected as FETAL BS.
9. While giving concentrations please place numbers first and the agent afterwards (for ex: %7 FBS, 1% penicillin streptomycin instead of FBS %7, penicillin-streptomycin %1).
10. Authors are suggested to use terms of real-time PCR or RT-qPCR instead of RT-PCR as it commonly refers to reverse transcriptase PCR.
11. A statistics section should be prepared.

Experimental design

I evaluated your manuscript for your in vitro experiments as other studies are not in my expertise. I have some minor inquiries about those studies but I also come up with some serious problems in your PCR analysis.
1. Please mention which method is used to quantify total RNA amounts.
2. Please mention the real time PCR device.
3. Please mention how many cycles were performed in real-time PCR.
4. Please make a table for your primer sequences and give information for base pairs and Tm values for the primers.
5. The RUNX2 primer is also binding with Cbfa1 gene in humans. So it’s not specific to RUNX2. Cbfa is also being effected by 17ß-estradiol and also an important gene in osteogenesis. So this is a major problem.
https://blast.ncbi.nlm.nih.gov/Blast.cgi?CMD=Get&RID=MBM9GR00013 (Link expires on 19.09.2021, 19:44 pm)
6. Your GAPDH primer is not specific for human GAPDH. It unfortunately makes your PCR results unusable.
https://blast.ncbi.nlm.nih.gov/Blast.cgi?CMD=Get&RID=MBMADSTF013 (Link expires on 19.09.2021, 19:45 pm)
7. Please mention cell seeding densities for MTT assay.

Validity of the findings

As the GAPDH primer is not showing specifity for human GAPDH gene and RUNX2 primer is also binds with Cbfa1 gene the PCR results are unfortunately unusable.

Cbfa1 gene expression is also being effected by 17ß-estradiol and have a role in osteogenesis as well. If the PCR primer binds with both RUNX2 and Cbfa1 we are actually given Ct values for both genes.

Housekeeping gene is being used for normalisation of the Ct values of other target genes. So if the housekeeping gene is not specific, both delta Ct values, delta delta Ct values and fold expression values will be wrong.

I have to stress that the manuscript is not publishable with these PCR results.

Additional comments

Dear Authors;
I had finished reviewing your manuscript on behalf of PeerJ. I unfortunately should inform you that I decided to suggest rejection to your manuscript due to non-specific PCR primers. Those results are unfortunately unusable as both your primer for the housekeeping gene is not specific and RUNX2 is also matches with Cbfa1 gene. Authors suggested to repeat PCR analysis with different PCR primers. My other comments are also visible in my detailed report at the end of the conclusions section.
Best Regards.

Annotated reviews are not available for download in order to protect the identity of reviewers who chose to remain anonymous.

Reviewer 2 ·

Basic reporting

The authors have done considerable research to explore the mechanism of heterotopic ossification and initially elaborated that 17-beta-estradiol may bind MKX and affect its interaction with SIN3A/HDAC1.However, the article contains many grammatical errors and repeated words, which may confuse the reader.I suggest the article be revised and published.

Experimental design

The article is scientific and innovative.

Validity of the findings

Conclusions are well stated, linked to original research question & limited to supporting results.

Additional comments

First, I recommend that authors carefully review their articles, remove repetitive words, and consult with native English speakers to embellish the article. Please avoid wordiness in academic writing and maintain concise description.All aspects of the study that have already been completed should be mentioned in past tense.
Below I list in detail the entries in the article that need to be changed for the author's reference.
① “Passaged BM-MSCs (n=1,000) were placed in each well of a 96-well plate, divided into three groups of 250 cells each, and were exposed to different doses of 17-beta-estradiol (0, 0.001, 0.01 nmol/L). ”This adds to 750, whereas the total number is 1,000. Were the remaining 250 considered the control group? You will have to state this clearly.
② “the expression of genes of cytokine RUNX2” RUNX2 is not a cytokine. ① Please rewrite this sentence to correct the description.
③ “RUNX2 primer forward: GGTGAAACTCTTGCCTCGTC, reversed: AGTCCCAACTTCCTGTGCT。GAPDH primer forward: GGCTGCCCAGAACATCAT, reversed: ATGATGTTCTGGGCAGCC。” The description and punctuations used are confusion. Please revise it.
④ “the method described in previous studies (Jiang et al., 2020; Zhao et al., 2021, p. 5)” You have mentioned the page number here, whereas you have not in other citations. Please review and delete this.
⑤ “A multifunctional microplate reader was used to measure OD values.” Such abbreviations need to be explained specifically as the reader—even if he/she is an expert in the field would expect clarity.
⑥ Since the structure of MKX is simulated, the conclusion lacks credibility. Please provide specific data for the prediction information.
⑦ Please elaborate on the relationship between the 5 substances ( Pro-Tyr, 17-beta-Estradiol, MKX, SIN3A/HDAC1).

Reviewer 3 ·

Basic reporting

This study is properly planned but needs to be improved in several ways, such as-


Grammatical issues.

experimental planining

Experimental design

In the experimental planning, some of the things overlooked such as-
no information is provided about the homology modeling i.e.,
1. Template chosen for modeling studies.
2. Before docking study, the structure was validated or not and the structure minimization.
3. Docking studies were performed by AutoDock Vina Software, but not validated by redocking analysis.
4. The results obtained from the Docking analysis should be included in the table format.
4. The complex obtained by molecular docking was further needed to be analyzed by MD simulation studies and the results should be included in the paper.

Validity of the findings

the above mentioned queries needs to be addressed to further enrich the manuscript.

Additional comments

NA

---

## Round 0.2 · accepted · Accept

Dear Authors,

After collecting the external comments, I am glad to inform you that the submission titled " Identification of the occurrence and potential mechanisms of heterotopic ossification associated with 17-beta-estradiol targeting MKX by bioinformatics analysis and cellular experiments" is accepted for publishing on PeerJ.

Reviewer 2 ·

Basic reporting

These additions and corrections have been made. With these modifications, I recommend publication.

Experimental design

no comment

Validity of the findings

no comment

Additional comments

no comment